# Pyrolysis Characteristics and Non-Isothermal Kinetics of Integrated Circuits

**DOI:** 10.3390/ma15134460

**Published:** 2022-06-24

**Authors:** Ziwei Chen, Linhao Liu, Hao Wang, Lili Liu, Xidong Wang

**Affiliations:** 1Department of Energy and Resources Engineering, College of Engineering, Peking University, Beijing 100871, China; zwchen@pku.edu.cn (Z.C.); liu-0806@163.com (L.L.); 2Beijing Key Laboratory for Solid Waste Utilization and Management, Peking University, Beijing 100871, China; 3Energy Bureau of Guangdong Province, Guangzhou 510030, China; a850159279@163.com; 4School of Energy and Environmental Engineering, University of Science & Technology Beijing, Beijing 100083, China; wanghao3352@163.com; 5School of Materials Science and Engineering, Peking University, Beijing 100871, China

**Keywords:** waste integrated circuits, pyrolysis characteristics, non-isothermal kinetics, electronic waste

## Abstract

Due to the complexity of components and high hazard of emissions, thermochemical conversions of plastics among waste-integrated circuits (ICs) are more favorable compared with the common treatment options of electronic waste (E-waste), such as chemical treatment and burning. In this study, the waste random-access memory, as the representative IC, was used to investigate the thermal degradation behaviors of this type of E-waste, including a quantitative analysis of pyrolysis characteristics and non-isothermal kinetics. The results show that the pyrolysis of the ICs can be divided into three different decomposition stages. The pyrolysis temperature and gas atmosphere play an important role in the pyrolysis reaction, and the heating rate greatly affects the rate of the pyrolysis reaction. The non-isothermal kinetic parameters and reaction mechanisms of ICs are determined using the Friedman method, Coats and Redfern (CR) method, and Kissinger method. The results show that the actual average activation energy of the pyrolysis reaction of ICs should be between 170 and 200 kJ·mol^−1^. The optimally fitting model for the ICs pyrolysis is the three-step parallel model consisting of the random nucleation model (A_m_) and reaction order model (C_n_).

## 1. Introduction

Over the past few decades, the electronic industries have been pushing factors of the modern industry’s advancement. With rapid technological advancement and increasing customer demand, the use of consumer electronic items has increased dramatically, and the products are becoming obsolete much faster than ever before [1,2]. Approximately 45 million tons of electronic waste (E-waste) were produced in 2012 around the world [3,4] and according to UN statistics, in 2018, there were 48.5 million tons of E-waste worldwide. E-waste is growing rapidly [5].

Even though a great deal of effort has been directed toward the promotion of harmless recycling of E-waste [6,7,8], recycling E-waste without harming the environment and human health is still a problem. E-waste generated by developed countries is usually exported to developing countries, where recycling techniques include burning and dissolution in strong acids with few measures to protect human health and the environment [9,10]. Over the past few decades, developing countries, especially China and India, have recycled the most E-waste worldwide at the expense of the environment and the health of local residents [11,12]. The amount of research on recycling E-waste without harming the environment and human health is not enough, and more research is required on the recycling technology of E-waste.

Integrated circuits (ICs), which play a vital role in electronic products as processors or memorizers, are widely used in electronic products such as computers, mobile phones, and tablets [3]. Currently, the research of artificial intelligence (AI) is attracting more and more attention, and ICs are one of the core parts of artificial intelligence. ICs will be more widely used with the development of AI. Little research has been conducted on the recycling of waste ICs [3,13]. On the one hand, the treatment of ICs is seldom concerned due to their long-life cycle. ICs have a longer life cycle than most electronic products. They can be used for eight years or more in normal situations. When electronic products are discarded, the ICs in them usually flow into the second-hand market. Therefore, the recycling of ICs has been ignored for a long time, which leads to the waste of resources. On the other hand, waste ICs are often mixed with other electronic wastes and burned together in the open air, and then precious metals are extracted chemically. The residue treated by this method not only wastes resources, but also produces many toxic chemicals such as flame retardants (PBDEs), dioxins/furans (PCDD/Fs), polycyclic aromatic hydrocarbons (PAHs), polychlorinated biphenyls (PCBs), and heavy metals. These toxic chemicals can highly pollute the environment and have a serious impact on the health of local residents [10,11,12]. Meanwhile, ICs have a high recycling value due to their high-value contents, especially precious metals [13]. The result of gold element analysis shows that the gold content of ICs is 534.3 g/t, as shown in Table 1, while the gold content of gold mines is 2~6 g/t generally. Therefore, it is necessary to study the recovery of ICs.

The recovery of ICs must consider not only the recovery of resources but also the environmental impact of the process. ICs are encapsulated by epoxy resin with added PBDEs. Serious environmental problems will be caused by improper recycling processes [14,15,16]. The following E-waste treatment options can be used for reference. Common E-waste treatment options are burning [17,18], chemical treatment [7,19,20], and mechanical separation [6,8,21,22]. Burning is the most common method to recycle E-waste. However, the process of treatment releases a large number of harmful gasses, such as polycyclic aromatic hydrocarbons (PAHs), hydrogen bromide (HBr), and even dioxins bromide. Burning residue can contaminate the soil and groundwater near the incineration site. Emissions of harmful gas and burning residue pose serious health risks to local residents [11,12,15]. Chemical treatment is efficient and relatively clean. However, a large amount of waste residue and liquid that are difficult to handle are caused by chemical treatment [7]. Lastly, mechanical separation is a good method in theory, but in fact, mechanical separation can only be used for the primary separation of the metal fraction and non-metal fraction from E-waste. The high local temperature of E-waste caused by mechanical separation can also produce harmful gases [6].

Supercritical fluid extraction [23,24,25] and pyrolysis [13,26,27,28] are promising methods for the harmless treatment of E-waste. Pyrolysis refers to the thermochemical decomposition of carbonaceous organic material in the absence or lack of oxygen through indirect heating to produce valuable fuels, such as pyrolysis gas, pyrolysis oil, and carbon black [28]. Wang and Zhang investigated the degradation process of brominated flame retardant (BFR) and BFR-containing waste computer housing plastic in various supercritical fluids (water, methanol, isopropanol, and acetone). Li and Xu used supercritical water (SCW) to decompose BER and recover metals from Waste Memory Modules (WMMs). Niu et al. used SCW to decompose the organics and recover Ta from Waste tantalum capacitors (WTCs). However, the technology of supercritical fluid extraction has strict requirements on the material of the equipment. More importantly, it is very expensive. Pyrolysis is a promising thermal technique, which can recover and remove organic material for resource recovery effectively and environmentally friendly. Diao et al. performed a series of molecular dynamics simulations (MDSs) to study the thermal decomposition characteristics of epoxy resin. Chen et al. researched vacuum pyrolysis characteristics and kinetic analysis of liquid crystal from scrap liquid crystal display panels. Liu et al. applied vacuum pyrolysis to degrade the organic material of waste ICs. Pyrolysis is considered a better treatment option than other options. Pyrolysis has the advantages of low environmental costs and high efficiency [13].

Little research has been conducted on the pyrolysis of waste ICs [13]. For this reason, the present study aims to investigate the thermal degradation characteristics and non-isothermal pyrolysis kinetics of ICs. Thermogravimetric analysis (TGA) is used to analyze the pyrolysis characteristics of organic material in Ics in a non-isothermal environment. The influence of pyrolysis-related factors on pyrolysis is studied. The non-isothermal kinetic parameters and kinetic reaction mechanisms of Ics are obtained by the Friedman method, Coats and Redfern (CR) method, and Kissinger method. The results of this study can provide process parameters and guidance for further industrial application of Ics pyrolysis technology to recycle ICs efficiently and environmentally friendly.

## 2. Materials and Methods

### 2.1. Materials

In this study, random-access memory (RAM), which was a representative IC, was used for the experiment. The RAM was selected as the sample for the following reasons. Firstly, the compositional characteristics of RAM are representative, and the pyrolysis characteristics of RAM can provide references for other types of ICs. Secondly, the application of RAM is widespread, and RAM generally has a higher content of precious metals and therefore has a higher recycling value than other types of ICs, such as a central processing unit, external memory, and so on. The samples were provided by the Zhongming Environmental Protection Group, which is an environmental company specializing in hazardous waste disposal and recycling. Prior to experiments, the materials were cut into small pieces by a shearing machine, and then ground into powder by a ball mill. The homogeneity of experimental samples can be guaranteed by turning the RAM into powder.

The composition of samples was determined by an Elemental Analyzer (Vario EL, Elementar Analysensysteme GmbH, Germany), X-Ray Fluorescence Spectrometer (Zetium, PANalytical B.V., Holland), Inductively Coupled Plasma-Atomic Emission Spectrometer (Prodigy 7, Leeman, America), and High-Performance Liquid Chromatography (HP1100, Agilent, America). The main components of ICs are listed in Table 1. Cu, C, Br, Si, and O are the main elements of ICs, accounting for more than 80 % of the components of ICs.

### 2.2. Experimental Methods

The pyrolysis experiments were performed in a tubular furnace on a laboratory scale. In order to control the atmosphere of the pyrolysis experiment, TGA was employed to deal with harmful waste gas, and a gas control system, an analytical balance system, and a tail gas treatment system were added, as shown in Figure 1. The analytical balance system consisted of an analytical balance, data line, and computer. During the experiment, the change in sample quality with time was transmitted to the computer through the data line. The change in temperature with time in the experiment was realized by measuring the temperature in the furnace with a thermocouple.

During the experiment, the sample was put into the crucible, and then the crucible was lifted by stainless-steel wire with oxidation resistance and corrosion resistance. The other end of the stainless-steel wire was connected to the analytical balance. The crucible was located in the middle of the quartz tube of a tubular furnace. The temperature of the experiment was controlled according to the setting procedure of the control panel of the tubular furnace. At the same time, the gas pressure relief valve and the gas flow control valve were opened to control the atmosphere of the pyrolysis experiment. Then the vacuum pump was opened to control the gas to pass through the tail gas treatment system. The airtightness of the whole experimental device was guaranteed.

### 2.3. Thermogravimetric Analysis

The sample quantity measured by the thermogravimetric analyzer was 2~5 mg, which is suitable for the thermogravimetric analysis of the pure substance. The relative error among the measurements of the thermogravimetric analyzer was approximately 3~5% [29,30]. While the powder sample made from the RAM was a mixture with an uneven composition, the relative error would be greater if the thermogravimetric analyzer was directly used for thermogravimetric analysis. So, the above experimental equipment was designed for thermogravimetric analysis of samples, as is shown in Figure 1.

For each experimental run, approximately 5 g of the sample was used. The samples were heated from 50 to 1050 °C at a constant heating rate of 10 °C/min. The system pressure is atmospheric pressure, and the gas atmosphere during pyrolysis is nitrogen and air. The flow rate of the carrier gas was fixed at 0.5 L/min. The relative error was controlled to less than 1%. In order to investigate the factors affecting the pyrolysis of ICs, the independent variables and control variables are shown in Table 2. Approximately 10 g of waste ICs were used in each experiment. The system pressure was one atmospheric pressure, and the flow rate of the carrier gas was fixed at 0.5 L/min. Samples before and after pyrolysis were analyzed by elemental analysis (C, H, N) to better understand the influence of different pyrolysis factors on the pyrolysis of ICs. Under any given condition, the experiment was usually carried out more than twice.

### 2.4. Pyrolysis Kinetics

The nomenclature used is shown in Table 3. The reaction equation of the pyrolysis of ICs can be written as follows:
(1)ICs (s)→B (s)+(l)+D (g)

The non-isothermal kinetics for the pyrolysis of ICs is written as follows [30]:(2)$$\frac{d\alpha }{\mathrm{dt}}=\mathrm{kf}\left(\alpha \right)$$
where α is the conversation rate and is given by
(3)$$\alpha =\frac{m_{i}-m}{m_{i}-m_{f}}$$

In the above equation, m, $m_{i}$, and $m_{f}$ represent the instantaneous, initial (at 50 °C), and final (at 1050 °C) weights of the sample. f(α) is the mechanical function of the reaction kinetics model and can be written as
(4)$$f\left(\alpha \right)={\left(1-\alpha \right)}^{n}$$

Common reaction kinetics models and mechanical functions are summarized in Appendix A. The reaction rate constant k is expressed in terms of the Arrhenius equation as:(5)$$k={\mathrm{Ae}}^{-\frac{E_{a}}{\mathrm{RT}}}$$
where A is the pre-exponential factor, ${\;E}_{a}$ is the activation energy, R is the universal gas constant, T is the absolute temperature, n is the reaction order. For a constant heating rate β:(6)$$\beta =\frac{\mathrm{dT}}{\mathrm{dt}}$$

Substituting Equations (4)–(6) into Equation (2) gives:(7)$$\frac{d\alpha }{\mathrm{dT}}=\frac{1}{\beta }{\mathrm{Ae}}^{-\frac{E_{a}}{\mathrm{RT}}}{\left(1-\alpha \right)}^{n}$$

Different methods, namely the Friedman method of Equation (8), CR method of Equations (9) and (10), and Kissinger method of Equation (11), are used in this work, as shown in Equations (8)–(11) [29,30].
(8)$$\ln(\frac{\beta_{i}}{T^{B}})=\mathrm{Const}-C\frac{E_{a}}{{\mathrm{RT}}_{\alpha }}$$
(9)$$\ln(\frac{-\ln\left(1-\alpha \right)}{T^{2}})=\ln\frac{\mathrm{AR}}{{\beta E}_{a}}-\frac{E_{a}}{\mathrm{RT}}\;\mathrm{if}\;n=1$$
(10)$$\ln(\frac{-\ln{\left(1-\alpha \right)}^{1-n}}{\left(1-n\right)T^{2}})=\ln\frac{\mathrm{AR}}{{\beta E}_{a}}-\frac{E_{a}}{\mathrm{RT}}\;\mathrm{if}\;n\neq 1$$
(11)$$\ln(\frac{\beta }{T_{m,i}^{2}})=\ln\left[-\frac{\mathrm{AR}}{E_{a}}\;f\prime \left(\alpha_{m}\right)\right]-\frac{E_{a}}{{\mathrm{RT}}_{m,i}}\;\mathrm{if}\;n\neq 1$$

## 3. Results and Discussion

### 3.1. Pyrolysis Properties of ICs

There are few direct studies on the pyrolysis properties of ICs [13], but there are many studies on the pyrolysis properties of waste PCBs, which are similar to those of ICs [5,14,18,31,32]. These studies and the component analysis of ICs can provide some basic information about their behavior as well as potential products and applications in their thermochemical conversion processes.

In the pyrolysis processes, carbon and hydrogen are the main reaction elements, as the organic components of ICs are mainly composed of carbon and hydrogen. Organic components in ICs will be decomposed into small organic molecules and released by heating. Carbon monoxide and carbon dioxide will also be generated in an air atmosphere. The pyrolysis of elemental nitrogen under an air atmosphere can produce nitrogen oxides (NO_x_). Unlike waste PCBs, ICs do not contain chlorine, and the bromine content is relatively high (9.17 wt%). Pyrolysis will produce bromine-based toxic gases, including hydrogen bromide (HBr), polybrominated biphenyls (PBBs), and even bromide dioxin. This requires methods to reduce the emissions of these toxic gases, and more research is needed on the migration and transformation of bromine in ICs. Copper, tin, and other metal elements are oxidized under the air atmosphere, but not under the nitrogen atmosphere. Noble metal elements will not react under any gas atmosphere.

Pyrolysis residue under the air atmosphere is mainly composed of metal oxides and glass cloth. Pyrolysis residue under a nitrogen atmosphere consists of metal, inorganic carbon, and glass cloth. The method of separating the glass cloth from the pyrolysis residue is the key to the recovery of ICs.

### 3.2. TG Analysis

The TGA and DTG curves of ICs under nitrogen and air atmospheres are shown in Figure 2. In general, the pyrolysis processes of ICs are characterized by a three-stage thermal degradation. In the first stage, the quality of the ICs changes negligibly. Some research shows there is PBDEs emission in the first stage [14,31]. The decomposition of ICs mainly takes place in the second stage. In this stage, the organic macromolecular polymer is thermally decomposed into a large number of volatile organic small molecules. In the third stage, as the temperature increases, the benzene ring begins to decompose.

The pyrolysis processes of ICs under **a** nitrogen atmosphere begin at 284 °C. The second-stage reactions develop at approximately 284~373 °C. The third-stage reactions develop at approximately 373~712 °C. The pyrolysis processes of ICs under the air atmosphere begin at 299 °C. The second-stage reactions develop at approximately 299~410 °C. The third-stage reactions develop at approximately 410~874 °C. After the pyrolysis reaction, there is a weight loss of 19.92 wt% for ICs under a nitrogen atmosphere and 19.54 wt% for ICs under an air atmosphere. The weight loss in pyrolysis of ICs is lower than the weight loss in pyrolysis of PCBs [13,32]. Because the organic content of ICs is usually lower than that of PCBs, the weight loss between ICs under nitrogen and ICs under air is similar. The results of elemental analysis before and after pyrolysis are shown in Figure 3. The pyrolysis residue of the reaction in nitrogen contains pyrolysis residual carbon, while the pyrolysis residue of the reaction in the air does not contain residual carbon, and the metal part of ICs will be oxidized by oxygen. From the similar pyrolysis mass loss, it can be concluded that the mass increase in the oxidized metal part is approximately equal to the mass of the pyrolysis residual carbon. The peak temperature of ICs under a nitrogen atmosphere (338 °C) is lower than that of ICs under an air atmosphere (358 °C), but the decomposition intensity of the former (−0.4529 wt.%·°C^−1^) is substantially higher than that of the latter (−0.2024 wt.%·°C^−1^). The pyrolysis reaction speed under the nitrogen condition is faster than that under the air condition. The pyrolysis reaction time under the nitrogen condition is shorter than that under the air condition. The initial temperature of the pyrolysis reaction under the nitrogen condition is lower than that under the air condition. The micro morphology of the samples before and after pyrolysis were analyzed by SEM, and the results are shown in Figure 3b–d. Before pyrolysis, the samples are mainly composed of metal particles and glass fibers, filled with organic materials. After pyrolysis under an N_2_ atmosphere, the organic materials are completely pyrolyzed when the metal particles change slightly. On the surface of the sample, there is a small amount of residual carbon. After pyrolysis under a nitrogen atmosphere, the surface of the sample becomes rougher, and the metal particles are coated with an oxide film. Therefore, compared with pyrolysis under an air atmosphere, pyrolysis under a nitrogen atmosphere can preserve the original state of the metal parts in the waste ICs, which is conducive to further refining for metal recovery.

### 3.3. Effect of Related Factors on Pyrolysis

The temperature, heating rate, and gas atmosphere are important factors affecting pyrolysis. In this study, the effect of temperature, heating rate, and gas atmosphere on the pyrolysis progress and products were studied. The results show that pyrolysis temperature and gas atmosphere have an important influence on the pyrolysis reaction, and the heating rate only affects the rate of the pyrolysis reaction and has no effect on the reaction results.

#### 3.3.1. Effect of Temperature

In order to investigate the effect of temperature on the pyrolysis of ICs, constant-temperature pyrolysis experiments were carried out at 250, 400, 600, and 800 °C. In the experiment, the tube heater is heated to a specified temperature, and then the sample is suspended in the tube heater. The selection of temperature-independent variables is based on the TGA and DTG curves of ICs under a nitrogen atmosphere.

The results are shown in Figure 4. At 250 °C, there is a considerable decrease in quality, although the rate of change is relatively slow. The PBDEs emission can explain this change [14,31]. At 400 °C, the reaction in the second stage of pyrolysis occurs. The organic macromolecular polymer is thermally decomposed into a large number of volatile organic small molecules. At the same time, most aromatic compounds containing benzene rings formed by pyrolysis are also released. Most organic components in ICs will be emitted at 400 °C. However, there are still a few organic compounds that do not react from elemental analysis of pyrolysis residue. The reaction rate of the second stage at 600 °C is quicker than that at 400 °C. Additionally, the reaction of the third stage can occur at 600 °C, but the reaction rate is slow. All organic components become residual carbon and volatile organic compounds from elemental analysis of pyrolysis residue. At 800 °C, the result of pyrolysis is similar. The reaction rate of the second stage is also similar compared to that at 600 °C. Nevertheless, the reaction rate in the third stage of pyrolysis at 800 °C is much faster than that at 600 °C. In conclusion, pyrolysis temperature has an important influence on the pyrolysis reaction. It affects the final composition of reactants and reaction rate and can be controlled to obtain different pyrolysis results.

Figure 4c–f shows the micro morphology of the samples before and after pyrolysis at different temperatures. For pyrolysis at 250 °C, a preliminary melting phenomenon appeared on the surface of the sample. Compared with pyrolysis at 250 °C, the surface reaction of samples pyrolyzed at 400 °C was more complete, and most of the organic components also reacted. After pyrolysis at 800 °C, the sample almost completely melted, and the organic components were fully reacted. However, the surface of the residue was seriously damaged. The higher the temperature, the more serious the damage to the part of the RAM that cannot be pyrolyzed. Hence, an appropriate temperature of 400°C is suggested. Under this condition, the rate of reaction can be guaranteed while protecting the metal part from destruction.

#### 3.3.2. Effect of Heating Rate

Heating rate is an important factor affecting the pyrolysis reaction, and 5 k/min, 10 k/min, and 15 k/min are set to study the effect of heating rate on pyrolysis of ICs. The results are shown in Figure 5.

At the heating rate of 5 k/min, the reaction time is the longest, while at the heating rate of 20 k/min, the reaction time is the shortest. The heating rate has a great influence on the reaction time of pyrolysis of ICs. The temperature at which the pyrolysis begins does not vary with the heating rate. Obviously, the heating rate has little influence on the reaction rate of the second stage of pyrolysis. This means that the reaction rate of the second stage is affected very little by the heating rate. The reaction rate of the third stage of pyrolysis increases as the heating rate increases. This result reflects that the reaction rate of the third stage is greatly influenced by the heating rate. The reaction time of the third stage can be shortened by increasing the heating rate of the third stage in future industrial applications. The results of elemental analysis of pyrolysis residue at different heating rates indicate the heating rate does not influence the final composition of reactants. In conclusion, the heating rate affects the rate of pyrolysis reaction. The desired reaction time can be obtained by controlling the heating rate, and the heating rate will not affect the results of the pyrolysis reaction.

#### 3.3.3. Effect of Gas Atmosphere

Pyrolysis experiments in the different gas atmospheres often mean different reaction processes. It is important to study the influence of the gas atmosphere on pyrolysis experiments. The independent variables and control variables considering the influence of the gas atmosphere on the pyrolysis of ICs are shown in Table 2. In order to study the effect of the different gas atmospheres on the pyrolysis reaction at different temperatures, the temperature is added as another independent variable.

The results are shown in Figure 6. The pyrolysis of ICs at different temperatures under nitrogen is different from the pyrolysis of ICs at different temperatures under air. At 250 °C, the reaction rate in air is quicker than that in nitrogen. There is a weight loss of 7.04 wt.% for ICs under nitrogen and 9.8 wt.% for ICs under air. The difference in weight loss can be explained by the difference in the carbon content of pyrolysis residue, and the pyrolysis residue under air has a lower carbon content (8.42 wt.%) compared with the carbon content (12.84 wt.%) of pyrolysis residue under nitrogen. Whether ICs have a pyrolysis reaction at 250 °C under air needs further study. At 400 °C, the reaction rate in air is slower than that in nitrogen. From the results of elemental analysis, the residue under air contains little carbon (0.54 wt.%). However, residue under nitrogen has an 8.41 wt.% carbon content. At 800 °C, the reaction rate is almost the same between air and nitrogen. Pyrolysis residue under air does not have carbon and pyrolysis residue under nitrogen also contains a considerable amount of carbon (6.43 wt.%).

In summary, the gas atmosphere has little effect on the pyrolysis reaction rate. The pyrolysis of ICs in different atmospheres has different reaction processes and results. The differences in pyrolysis reaction at different atmospheres need further investigation.

### 3.4. Non-Isothermal Kinetics of ICs

#### 3.4.1. Reaction Activation Energy

The experimental data of the pyrolysis of ICs under a nitrogen atmosphere at different heating rates were used to analyze the non-isothermal kinetics. Figure 7a shows the relationship between $\ln\left(\frac{d\alpha }{\mathrm{dt}}\right)$ and $\frac{1}{T}$ obtained by the Friedman method. When α is less than 0.65, the linear correlation between $\ln\left(\frac{d\alpha }{\mathrm{dt}}\right)$ and $\frac{1}{T}$ is good, and the slope changes very little, indicating that the activation energy does not change much. When α is greater than 0.65, the linear correlation between A and B is low, and the slope of the dotted line becomes larger, which means that the activation energy becomes larger, and the accuracy of this method decreases. This shows the complexity of the pyrolysis reaction. When the reaction conversion rate is low, the pyrolysis reaction is dominated by one type of reaction, and the activation energy of the reaction varies slightly. When the reaction conversion rate is high, the pyrolysis reaction is jointly dominated by various types of reactions and thus the activation energy becomes larger. In addition, the drastic changes in the activation energy of the reaction can be seen at both ends of the reaction conversion rate, indicating that this method amplifies the random error here. Figure 7b shows the relationship between $\ln\left(\frac{\beta }{T_{m}^{2}}\right)$ and $\frac{1}{T_{m}}$ obtained by the Kissinger method. The activation energy can be obtained from the slope and is 170.90 kJ·mol^−1^, which is close to the activation energy obtained by the Friedman equation when the reaction conversion rate is lower than 0.7. The pyrolysis kinetics model of the CR equation method is a first-order reaction. The relationship between $\ln(\frac{-\ln\left(1-\alpha \right)}{T^{2}})$ and $\frac{1}{T}$ is shown in Figure 7c. It can be seen that the relationship is not completely linear. The curve can be divided into two sections for linear fitting, and the results are shown in Appendix A. The linear correlation of the piecewise fitting is good, and R^2^ is above 0.9. It can be seen that the activation energy obtained by fitting in the first section is smaller than that obtained by the first two methods, while the activation energy obtained in the second section is very small, which deviated from the normal range of common organic pyrolysis activation energy. Therefore, the CR equation may not be suitable for the complex kinetics of the solid thermal decomposition reaction.

Table 4 shows the values of activation energy obtained by the three methods. It is worth noting that the adopted activation energy of the CR equation is the average of the activation energy of the first section. The reaction activation energy obtained by the CR method is relatively small, while that obtained by the Friedman method is relatively large. When considered comprehensively, the actual average activation energy of the pyrolysis reaction of the waste memory module should be between 170 and 200 kJ·mol^−1^. The obtained results on the activation energy correspond to those reported in the study of Jadhao et al. [33]. The comparison of activation energy in this study with the literature data [33,34,35,36] is displayed in detail in Appendix A.

#### 3.4.2. Kinetics Model for ICs Pyrolysis

The kinetics model for pyrolysis was analyzed using the Master plot method. As shown in Figure 8a–c, the corresponding g(α), (b) y(α), and z(α) curves were determined. It can be seen that the rising speed of the g(α) curve of the ICs pyrolysis firstly increases, then slows down, and finally accelerates, of which the characteristics are similar to that of random nucleation models A2 and A3. The front part of the y(α) curve of the ICs pyrolysis coincides with that of random nucleation models A2 and A3 on the whole. At the end, the y(α) curve decreases faster than that of the random nucleation model and falls between the reaction order curves F1 and C2. The z(α) curve of the ICs pyrolysis is also most similar to random nucleation models A2 and A3 in the front, and the end of the curve is close to the reaction order model C2. Therefore, it can be concluded that the process of ICs pyrolysis can be described by the hybrid dynamic model consisting of the random nucleation model and reaction order model. Based on the information, the model parameters can be calculated to verify the correctness of the model.

#### 3.4.3. Kinetic Parameters of ICs Pyrolysis

Based on the obtained information above, different parallel reaction models were developed, including the two-step parallel model A_m_-C_n_, three-step parallel model A_m_-C_n_-C_n_, three-step parallel model A_m_-C_n_-D_1_, and four-step parallel model A_m_-C_n_-C_n_-C_n_. For two, three, and four-step parallel models, the relationship between conversion rates can be described as follows
(12)$$\alpha =\sum_{i=1}^{i=n}w_{i}\alpha_{i}\;,\sum_{i=1}^{i=n}w_{i}=1$$
where n is the number of parallel reaction steps. Nonlinear fitting was conducted to obtain the model parameters. The fitting process and variances are shown in Figure 9a–e, and the obtained model parameters are listed in Table 5. On the whole, in the four models, the A_m_-C_n_-C_n_ model shows the optimal fitting performance with the root-sum square (RSS) of 0.0021, 0.00831, and 0.01682 at the heating rates of 5 K/min, 10 K/min, and 20 K/min, respectively. The comparison of fitting variances (S^2^) also confirms this result. Hence, the three-step parallel model A_m_-C_n_-C_n_ is the optimal fitting model for the ICs pyrolysis.

## 4. Conclusions

The pyrolysis characteristics of ICs were investigated in this study through thermogravimetric analyses. Three different decomposition stages during the pyrolysis of ICs were clearly identified. The pyrolysis reaction speed under the nitrogen condition is faster than that under the air condition. The pyrolysis reaction time under the nitrogen condition is shorter than that under the air condition. It is also worth mentioning that the gas cost of pyrolysis under nitrogen is higher than that under air. The influence of pyrolysis-related factors on the pyrolysis of ICs has been studied by the control variable method. Pyrolysis temperature affects the final composition of reactants and reaction rate and can be controlled to obtain different pyrolysis results. The heating rate has little influence on the reaction rate of the second stage of pyrolysis. The reaction rate of the third stage of pyrolysis increases as the heating rate increases. Different gas atmospheres, or the presence or absence of oxygen, lead to different reaction processes and results of pyrolysis of ICs. Non-isothermal kinetics of ICs has been examined through the Friedman method, CR method, and Kissinger method. The reaction activation energy obtained by the CR method is relatively small, while that obtained by the Friedman method is relatively large. Considered comprehensively, the actual average activation energy of the pyrolysis reaction of the waste memory module should be between 170 and 200 kJ·mol^−1^. The process of ICs pyrolysis can be described by the three-step parallel model (A_m_-C_n_-C_n_). Overall, an efficient and environmentally friendly process technology (pyrolysis) for organic material degradation in waste ICs was determined when pyrolysis gas is considered for use as a circulating fuel. This study provided thermodynamic data and kinetic parameters for further industrial applications.

## Figures and Tables

**Figure 1 materials-15-04460-f001:**
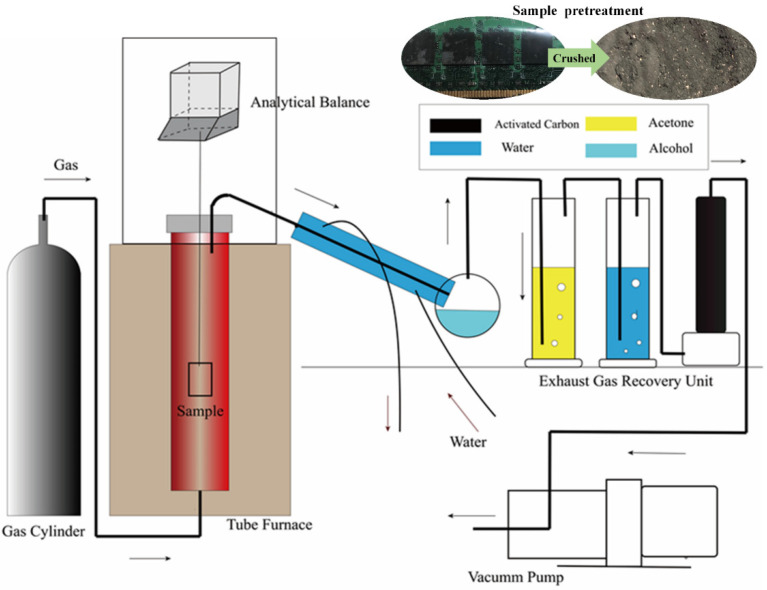
Schematic diagram of experimental device.

**Figure 2 materials-15-04460-f002:**
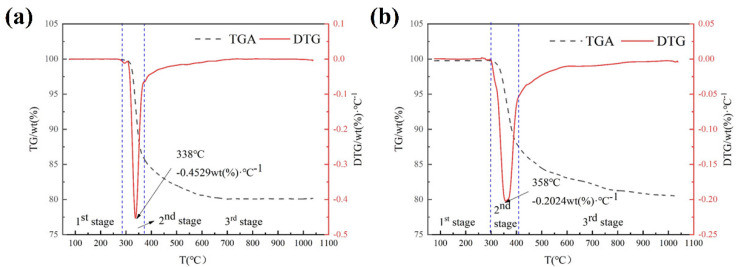
TGA and DTG curves of ICs under (**a**) nitrogen and (**b**) air atmospheres. The three pyrolysis stages are marked by blue lines.

**Figure 3 materials-15-04460-f003:**
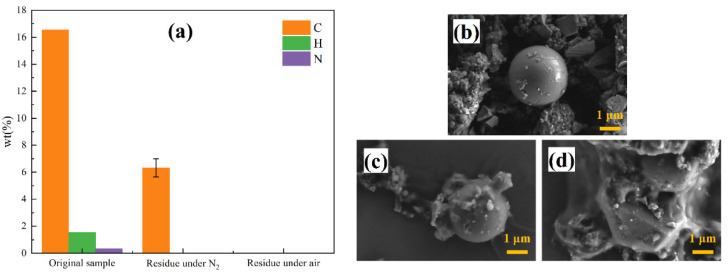
(**a**) Elemental analysis of samples before and after pyrolysis; (**b**) micro morphology of samples before pyrolysis; micro morphology of samples after pyrolysis under (**c**) an N_2_ atmosphere and (**d**) an air atmosphere, respectively.

**Figure 4 materials-15-04460-f004:**
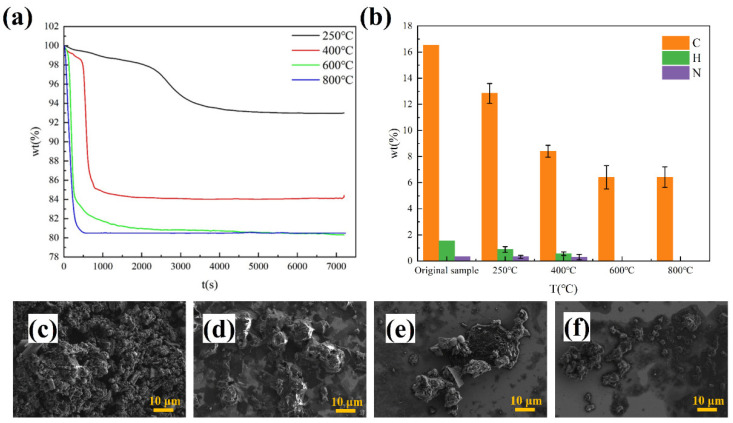
(**a**) Changes of mass with time at different temperatures; (**b**) elemental analysis of pyrolysis residue at different temperatures; micro morphology of samples (**c**) before pyrolysis and after pyrolysis at (**d**) 250 °C, (**e**) 400 °C, and (**f**) 800 °C.

**Figure 5 materials-15-04460-f005:**
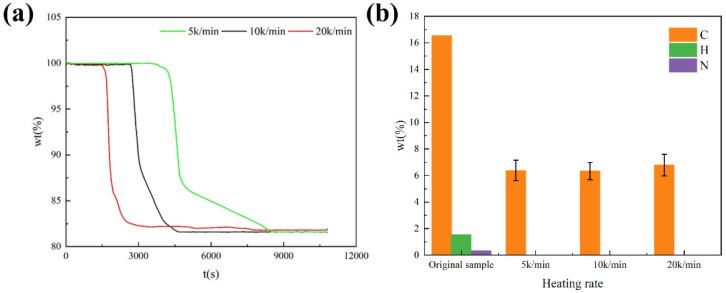
(**a**) Changes of mass with time at different heating rates; (**b**) elemental analysis of pyrolysis residue at different heating rates.

**Figure 6 materials-15-04460-f006:**
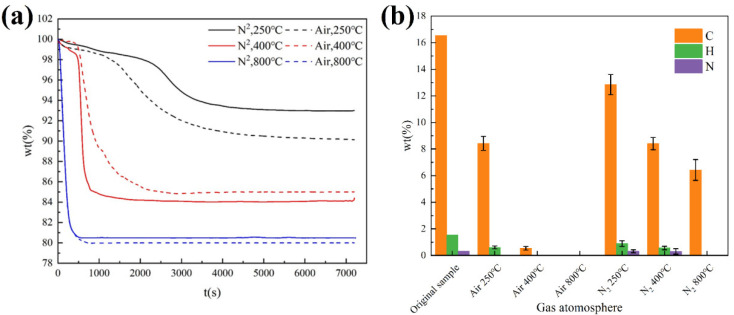
(**a**) Changes of mass with time at different gas atmospheres and temperatures; (**b**) elemental analysis of pyrolysis residue at different gas atmospheres and temperatures.

**Figure 7 materials-15-04460-f007:**
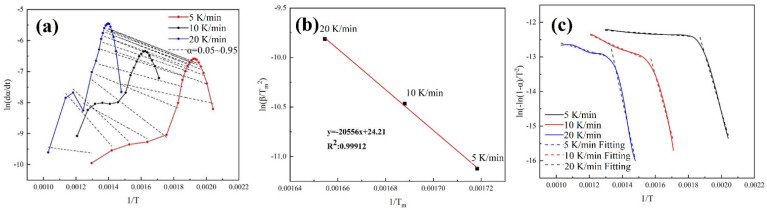
(**a**) The relationship between $\ln\left(\frac{d\alpha }{\mathrm{dt}}\right)$ and $\frac{1}{T}$ obtained from Friedman method; (**b**) the relationship between $\ln\left(\frac{\beta }{T_{m}^{2}}\right)$ and $\frac{1}{T_{m}}$ obtained from Kissinger method; (**c**) the relationship between $\ln(\frac{-\ln\left(1-\alpha \right)}{T^{2}})$ and $\frac{1}{T}$ obtained from CR method.

**Figure 8 materials-15-04460-f008:**
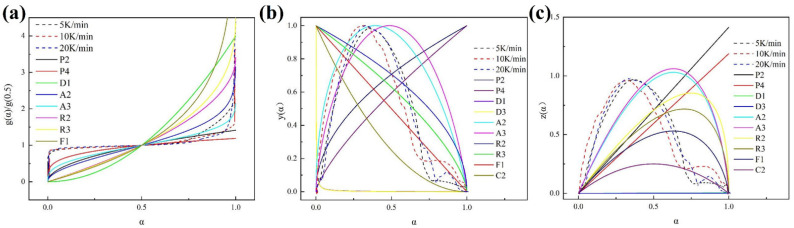
(**a**) g(α), (**b**) y(α), and (**c**) z(α) curves of pyrolysis of ICs and different standard kinetics models.

**Figure 9 materials-15-04460-f009:**
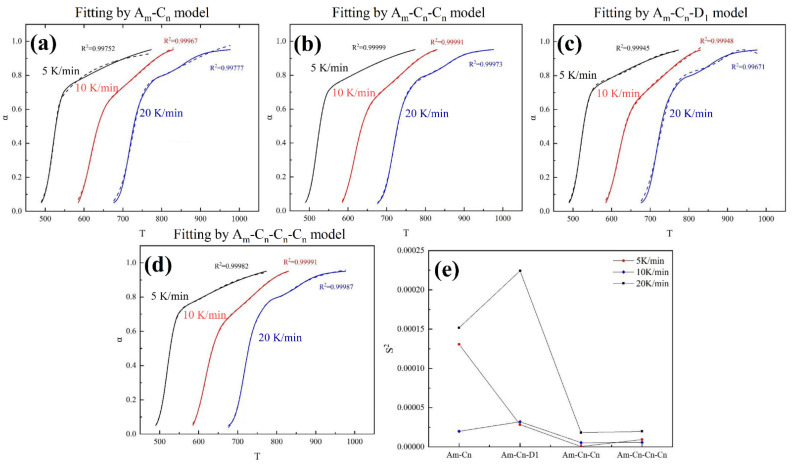
Fitting results by (**a**) two-step parallel model A_m_-C_n_, (**b**) three-step parallel model A_m_-C_n_-C_n_, (**c**) three-step parallel model A_m_-C_n_-D_1_, and (**d**) four-step parallel model A_m_-C_n_-C_n_-C_n_; (**e**) comparison of fitting variances of different parallel models.

**Table 1 materials-15-04460-t001:** Main compositions of ICs.

Main Compositions	Content (wt.%)	Precious Metal	Content (g/t)
Cu	17.82	Au	534.3
C	16.54	Ag	593
SiO_2_	33.29	Pt	0.78
Ca	4.72	Pd	5.91
Br	9.17	Rh	0.2
Sn	2.29		
Al	2.65		
Fe	1.63		
H	1.43		
N	0.33		
Pb	0.87		
Mg	0. 18		

**Table 2 materials-15-04460-t002:** Independent and control variables of experiments on the effect of pyrolysis-related factors.

Pyrolysis Related Factors	Independent Variables	Control Variables
Temperature	250 °C	Heat rate: 10 k/min Reaction time: 2 h Gas atmosphere: N_2_
	400 °C	
	600 °C	
	800 °C	
Heating rate	5 k/min	Temperature: 800 °C Reaction time: 3 h Gas atmosphere: N_2_
	10 k/min	
	20 k/min	
Gas atmosphere	250 °C, N_2_	Heat rate: 10 k/min Reaction time: 2 h
	250 °C, Air	
	400 °C, N_2_	
	400 °C, Air	
	600 °C, N_2_	
	600 °C, Air	

**Table 3 materials-15-04460-t003:** List of nomenclature.

Nomenclature	Full Name	Nomenclature	Full Name
A	Pre-exponential factor (min^−1^)	T	Temperature (K)
B	Heating rate (K·min^−1^)	t	Heating time (min)
E_a_	Activation energy (J·mol^−1^)	α	Conversion of sample
k	Rate constant (min^−1^)	m	Weight of sample (g)
n	Order of reaction	i	Initial state
R	Universal gas constant (J·mol^−1^·K^−1^)	f	Final state

**Table 4 materials-15-04460-t004:** Activation energy obtained from the Friedman method, CR method, and Kissinger method.

Methods	Friedman	Kissinger	CR
E_a_ (kJ·mol^−1^)	208.63	170.90	145.21

**Table 5 materials-15-04460-t005:** Fitting parameters of models.

Models	Heating Rate	w_i_	Parameters	f(α)	Root-Sum Square
A_m_-C_n_	5 K/min	w_1_ = 0.45 w_2_ = 0.55	A_1_ = 2.58 × 10^17^, E_1_ = 205.69 kJ/mol A_2_ = 1.35 × 10^13^, E_2_ = 152.45 kJ/mol	A_m_:m_1_ = 10.0 C_n_:m_2_ = 8.1	0.46157
	10 K/min	w_1_ = 0.40 w_2_ = 0.60	A_1_ = 3.63 × 10^17^, E_1_ = 285.62 kJ/mol A_2_ = 1.46 × 10^14^, E_2_ = 191.69 kJ/mol	A_m_:m_1_ = 0.2 C_n_:m_2_ = 1.8	0.03042
	20 K/min	w_1_ = 0.29 w_2_ = 0.71	A_1_ = 1.55 × 10^15^, E_1_ = 288.09 kJ/mol; A_2_ = 6.559 × 10^13^, E_2_ = 216.03 kJ/mol	A_m_:m_1_ = 0.2 C_n_:m_2_ = 1.4	0.14106
A_m_-C_n_-C_n_	5 K/min	w1 = 0.11 w2 = 0.24 w3 = 0.65	A_1_ = 7.2 × 10^16^, E_1_ = 219.54 kJ/mol A_2_ = 2.03 × 10^14^, E_2_ = 184.10 kJ/mol A_3_ = 1.97 × 10^15^, E_3_ = 174.24 kJ/mol	A_m_:m_1_ = 0.3 C_n_:m_2_ = 6.9 C_n_:m_3_ = 1.5	0.0021
	10 K/min	w1 = 0.12 w2 = 0.23 w3 = 0.65	A_1_ = 3.63 × 10^17^, E_1_ = 235.62 kJ/mol A_2_ = 3.46 × 10^14^, E_2_ = 191.69 kJ/mol A_3_ = 1.14 × 10^15^, E_3_ = 183.80 kJ/mol	A_m_:m_1_ = 0.4 C_n_:m_2_ = 6.1 C_n_:m_3_ = 1.8	0.00831
	20 K/min	w1 = 0.15 w2 = 0.22 w3 = 0.65	A_1_ = 4.97 × 10^17^, E_1_ = 248.09 kJ/mol; A_2_ = 2.44 × 10^14^, E_2_ = 210.15 kJ/mol A_3_ = 2.05 × 10^15^, E_3_ = 203.80 kJ/mol	A_m_:m_1_ = 0.5 C_n_:m_2_ = 6.1 C_n_:m_3_ = 1.5	0.01682
A_m_-C_n_-D_1_	5 K/min	w1 = 0.26 w2 = 0.73 w3 = 0.01	A_1_ = 3.38 × 10^20^, E_1_ = 302.65 kJ/mol A_2_ = 5.81 × 10^14^, E_2_ = 169.20 kJ/mol A_3_ = 7.15 × 10^10^, E_3_ = 146.12 kJ/mol	A_m_:m_1_ = 0.2 C_n_:m_2_ = 1.7	0.10047
	10 K/min	w1 = 0.38 w2 = 0.61 w3 = 0.07	A_1_ = 2.41 × 10^18^, E_1_ = 244.34 kJ/mol A_2_ = 5.42 × 10^13^, E_2_ = 194.99 kJ/mol A_3_ = 1.24 × 10^10^, E_3_ = 112.52 kJ/mol	A_m_:m_1_ = 0.7 C_n_:m_2_ = 8.2	0.04883
	20 K/min	w1 = 0.17 w2 = 0.82 w3 = 0.01	A_1_ = 4.95 × 10^20^, E_1_ = 385.22 kJ/mol; A_2_ = 2.62 × 10^13^, E_2_ = 210.24 kJ/mol A_3_ = 2.17 × 10^13^, E_3_ = 198.02 kJ/mol	A_m_:m_1_ = 0.5 C_n_:m_2_ = 1.9	0.20789
A_m_-C_n_-C_n_-C_n_	5 K/min	w1 = 0.48 w2 = 0.23 w3 = 0.23 w4 = 0.16	A_1_ = 2.72 × 10^18^, E_1_ = 198.26 kJ/mol A_2_ = 9.62 × 10^14^, E_2_ = 176.53 kJ/mol A_3_ = 9.6 × 10^14^, E_3_ = 176.53 kJ/mol A_4_ = 9.84 × 10^14^, E_4_ = 179.06 kJ/mo	A_m_:m_1_ = 0.8 C_n_:m_2_ = 5.1 C_n_:m_3_ = 5.1 C_n_:m_4_ = 4.2	0.03323
	10 K/min	w1 = 0.19 w2 = 0.28 w3 = 0.28 w4 = 0.25	A_1_ = 9.12 × 10^40^, E_1_ = 500.00 kJ/mol A_2_ = 1.68 × 10^14^, E_2_ = 194.19 kJ/mol A_3_ = 1.66 ×10^14^, E_3_ = 194.16 kJ/mol A_4_ = 3.85 × 10^12^, E_4_ = 211.72 kJ/mol	A_m_:m_1_ = 0.4 C_n_:m_2_ = 2.7 C_n_:m_3_ = 2.8 C_n_:m_4_ = 3.7	0.00875
	20 K/min	w1 = 0.46 w2 = 0.24 w3 = 0.24 w4 = 0.16	A_1_ = 1.99 × 10^15^, E_1_ = 237.60 kJ/mol; A_2_ = 3.48 × 10^15^, E_2_ = 252.56 kJ/mol A_3_ = 3.48 × 10^15^, E_3_ = 252.56 kJ/mol A_4_ = 5.54 × 10^16^, E_4_ = 285.47 kJ/mol	A_m_:m_1_ = 1.1 C_n_:m_2_ = 5.0 C_n_:m_3_ = 5.0 C_n_:m_4_ = 1.4	0.00829

## Data Availability

The data presented in this study are available on request from the corresponding author.

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
