# Peer review of "Pyrolysis Characteristics and Non-Isothermal Kinetics of Integrated Circuits"

_materials, 2022, doi:10.3390/ma15134460_

Round 1
Reviewer 1 Report
In this paper, the authors presented an experimental investigation of pyrolysis of ICs via thermogravimetric analyses.
Ram was employed as feedstocks.
A parametric study involving the effects of temperature, heating rate, and gas atmosphere (air vs. N2) on mass changes and residue mass composition was presented.
Overall, this review paper is well written and well organized. The details of the paper are complete, composed of the theory of pyrolysis, physical and chemical properties, the effect of operating parameters, etc.
I recommend accepting this work after minor revision.
The comments are stated below:
- line 51, suppress the bracket of AI
- line 110, explain more details in your manuscript why the random-access memory (RAM) was selected for this study and please add the photograph of this feedstock in the paper.
- line 161, the authors stated that “The experiment under any given condition was usually carried out more than twice” therefore please add the error bar in Figs 3, 4, 5, 6, 7, and 8 to demonstrate the repeatability and accuracy of your results.
- In the conclusion part, the authors say only the advantage of using N2 to pyro-reaction. The authors have to state the drawback of using N2 in terms of cost too.
- In the conclusion part, the value activation energy in the conclusion is required
- In the conclusion part, the authors stated that “Overall, an efficient and environmentally friendly process technology (pyrolysis) for organic material degradation in waste ICs was determined”. In fact, your process was not environmentally friendly as it emitted toxic gases. Please clarify how it is environmentally friendly; otherwise please use another adjustive.
Reviewer 2 Report
In my opinion the paper needs to be improved for publication.
1. Other papers carried out on the kinetics of pyrolysis and combustion of electronic wastes should be analyzed and compared.
2. The term pyrolysis in process is reserved for a non-oxidizing atmosphere (nitrogen, argon) while combustion, oxidative pyrolysis is carried out in the presence of air or a slightly or partially oxidizing atmosphere. These concepts should be reviewed at work.
3. The maximum decomposition range or the residue formed depending on the operating temperature should be clarified, and possibly more experiments with different temperatures are needed. The possible oxidation of metals must be analyzed.
4. The presented kinetic study raises many questions. It should be clarified if the conversion values ​​used are the global ones or the partial ones for each decomposition range. The parameters obtained are purely fitting-parameters, without any chemical significance, and should be compared with other values ​​in the literature. The decomposition model presented is incomplete and it cannot be used for interpolation/extrapolation of data and reactor design.
Round 2
Reviewer 2 Report
Although the authors have replied and introduced some changes, there is no a detail answer to the different questions. The decision of publication must be taken by the Editor. As the other reviewer consider that that paper can be published, I can accept the publication of the manuscript
